# FMCW Radar System Interference Mitigation Based on Time-Domain Signal Reconstruction

**DOI:** 10.3390/s23167113

**Published:** 2023-08-11

**Authors:** Zhengguang Xu, Shanyong Wei

**Affiliations:** School of Electronic Information and Communications, Huazhong University of Science and Technology, Wuhan 430074, China; weishanyong@hust.edu.cn

**Keywords:** frequency-modulated continuous-wave radar, time-domain signal reconstruction, interference detection, interference mitigation

## Abstract

In this study, an interference detection and mitigation method is proposed for frequency-modulated continuous-wave radar systems based on time-domain signal reconstruction. The interference detection method uses the difference in one-dimensional fast Fourier transform (1D-FFT) results between targets and interferences. In the 1D-FFT results, the target appears as a peak at the same frequency point for all chirps within one frame, whereas the interference appears as the absence of target peaks within the first or last few chirps within one frame or as a shift in the target peak position in different chirps. Then, the interference mitigation method reconstructs the interference signal in the time domain by the estimated parameter from the 1D-FFT results, so the interference signal can be removed from the time domain without affecting the target signal. The simulation results show that the proposed interference mitigation algorithm can reduce the amplitude of interference by about 25 dB. The experimental results show that the amplitude of interference is reduced by 20–25 dB, proving the effectiveness of the simulation results.

## 1. Introduction

Frequency-modulated continuous-wave (FMCW) radars have become a key sensor in advanced driver assistance systems (ADAS) due to their high resolution, all-weather capability, and simple system structure. With the increasing number of vehicles equipped with radar sensors on the road and limited spectrum resources, FMCW radar systems installed on different vehicles will inevitably be affected by interference from adjacent vehicle radar systems and other FMCW radars on the same vehicle. The main purpose of FMCW radars in preventing traffic accidents is collision avoidance and target detection. If interference occurs between radars and affects the normal operation of the radar system, it may cause ADAS to make incorrect judgments, resulting in incalculable losses. Therefore, it is critical to keep the vehicle radar system operational and mitigate mutual radar interference.

FMCW radars obtain specific information about a target using the intermediate frequency (IF) signal between the transmitted signal and the target reflection signal. However, if the interfered frequency difference signal is directly processed using two-dimensional fast Fourier transform (2D-FFT), the target detection performance will be poor. Specifically, some noise levels will be elevated, drowning out weak targets and reducing the target detection rate. The appearance of ghost targets causes radar false alarms. Over the past fifteen years, the properties and impacts of these interferences have been extensively studied [1,2]. The derivation of the interference duration and the form of the interference in an FMCW radar receiver due to different waveforms can be found in [3]. When the modulation slopes of two FMCW radar signals are different, it causes an elevation of the noise floor. In [4], the influence of interference on range-Doppler imaging is studied in this case. When the modulation slopes are the same, ghost targets may appear. The analytical formula for calculating the probability of ghost target occurrence was presented in [5]. The appearance of ghost targets is a more severe and critical anomaly for current ADAS systems compared to the elevation of the noise floor. The scenario of ghost target interference is shown in Figure 1. The vehicle in the picture is equipped with two forward-facing radars. When the signal transmitted by the left radar is reflected by the vehicle in front and received by the right radar, ghost target interference will form.

From 2010 to 2012, the European Union project More Safety for All by Radar Interference Mitigation (MOSARIM) [6] conducted extensive research on vehicle radar interference and divided interference mitigation (IM) methods into six categories: polarized antenna, time-domain, frequency-domain, spatial-domain, coding, and strategy coordination methods. The above IM methods are to perform corresponding signal processing at the transmitting or receiving signal. Among them, the methods in the polarized antenna and spatial-domain categories have a limitation in resolving interference issues in vehicle-mounted radars. Therefore, the methods in the time-domain, frequency-domain, coding, and strategy-domain categories are mainly considered to effectively solve the interference problem of vehicle-mounted FMCW radars.

Time domain: Find the location of the interference in the time domain, and then use the windowing method [7,8] to mitigate the interference. Additionally, an adaptive filtering method based on a phase-modulated continuous-wave radar system [9] and a method based on morphological component analysis [10] for IM have already been proposed elsewhere. In [11], a method is proposed to minimize the correlation between radar waveforms using the particle swarm optimization algorithm. The optimal filtering is achieved by the least mean squares (LMS) algorithm based on the reference signal to output interference suppression result. The main problem with time-domain methods is accurately locating the interference and then completely removing it from the signal.

Frequency domain: An adaptive noise cancellation IM method was proposed in [12]. A wavelet de-noising method [13] is also used to extract interference signals in the time domain and then suppress them to achieve IM. A two-stage approach for suppressing the mutual interference between FMCW radars was proposed in [14]. In the first stage, the signals corresponding to the strong interference components or targets are separated using the singular value decomposition (SVD) technique across the spatial domain. Following this, each separated signal at each receive channel is further decomposed into different frequency components using various mode decomposition techniques. In [15], IM is primarily achieved by cutting the interfered signal samples of FMCW radars in the short-time Fourier transform domain and then reconstructing the signal in the interfered area using an auto-regressive (AR) model based on the signal that has not been interfered with. In [16], a constant false alarm rate (CFAR)-based framework is proposed to mitigate interference in FMCW radars. In this framework, a one-dimensional CFAR detector is used for interference detection (ID), and signal reconstruction is performed using the Burgs method in the AR model or the amplitude correction method. Frequency-domain methods mainly identify the interference position in the frequency domain and then suppress the interference. However, to suppress the interference frequency as much as possible, the detected interference area is typically extended in the frequency domain, which can have a significant impact on the power of the useful signal.

Coding domain: In [17], an effective method for reducing mutual interference is proposed by performing random sub-band spectral analysis. The main implementation method is to randomly split a large triangular wave voltage into multiple triangular waves in the signal control unit. In [18], a new phase-coded FMCW system is proposed, where the transmitted waveform uses bipolar phase coding, and a group delay filter is used to process the unaligned phase-coded radar echo signals. Moreover, the orthogonality of randomly coded chirp signals [19] is used for IM. Coding-domain methods require changes in the radar operating mode or the style of the transmitted signal, they have high implementation complexity, and it is difficult to establish unified standards.

Strategy methods, such as reinforcement learning techniques [20,21], deep learning techniques [22,23], and generative adversarial networks [24], have been used for IM in FMCW radars. These methods require a large amount of data for training, they have high computational complexity, and the hardware implementation is complex.

At present, there are many studies on radar IM, but there are few studies on ghost target IM as shown in Figure 1. This article focuses on the detection and mitigation of this interference. In this study, we propose a framework for interference detection and mitigation in FMCW radars based on time-domain signal reconstruction. In the proposed framework, both the interference and reflected signals from the target are mixed with the transmission signal of the victim radar, which is identical to the transmission signal of the aggressor radar. Then, target detection is performed based on the 1D-FFT result of the IF signal. Furthermore, the range index where the interference occurs is found according to the target result. The interference signal frequency is calculated based on the 1D-FFT results of the interference point and its two adjacent points. In addition, the amplitude and phase of the interference signal are calculated based on the interference signal frequency. After the interference signal is recovered, it is removed from the original data samples to suppress the influence of ghost targets. Compared with existing methods, our method can effectively remove ghost targets without affecting the target signal. The interference detection and mitigation effect of our method have been validated through simulation and experimental results making it suitable for real-time interference detection and mitigation in FMCW radars. The main contributions of the study may be summarized as follows:

Many relevant studies focus more on IM methods and neglect ID methods. This article proposes a clear and feasible ID method to distinguish between normal targets and interferences.

This article proposes an IM method with strong suppression ability, which can achieve higher anti-interference performance and interference intensity reduction than other methods under high signal–noise ratio (SNR) conditions.

This article proposes a complete framework for an FMCW radar system, which includes three steps: target detection, interference detection, and interference mitigation.

The rest of this paper is organized as follows: Section 2 briefly introduces the signal model of FMCW radars. Then, the proposed interference detection and mitigation method based on time-domain signal reconstruction is presented in Section 3. Simulation and experimental results are presented in Section 4 and Section 5, respectively, to demonstrate the anti-interference performance of our proposed method. Finally, the conclusions and outlook are presented in Section 6.

## 2. Signal Model

Assume that the transmitted signal $s_{c}(t)$ of the victim radar A in one chirp is as follows:(1)$$s_{c}(t)=e^{j2\pi (f_{0}t+\frac{1}{2}St^{2}+\varphi )}0\leq t<T_{c}$$
where $f_{0}$ denotes the initial frequency of the transmitted signal of radar A, S denotes the modulation slope, φ denotes the initial phase of the transmitted signal of radar A, and $T_{c}$ denotes the duration of the chirp. As depicted in Figure 2, the transmitted signal s(t) of radar A in one frame is as follows:(2)$$s_{f}\left(t\right)=\left\{\begin{array}{cc} s_{c}\left(t-mT_{c}\right) & mT_{c}\leq t<\left(m+1\right)T_{c},m=0,1,\cdots ,M-1\\ 0 & MT_{c}\leq t<T_{f} \end{array}\right.$$
where M represents the number of chirps within a frame, m represents the number of chirps, and $T_{f}$ denotes the duration of a frame.

When the modulation slope of the signal transmitted by the aggressor radar is the same as that of the victim radar, the aggressor radar signal is received by radar A, and the interfering signal $s_{\mathrm{int}}(t)=s(t-\tau_{d}(t))$, where $\tau_{d}(t)$ is a time delay function, consists of delay and time drift. For the convenience of research, we assume that each frame of radar data is independent. When analyzing in one chirp, we consider the frequency difference between the received signal and interference signal to be approximately constant, neglecting the minor effects of time drift. Considering *P* multiple targets and *Q* multiple interfering signals simultaneously, the signal collected by radar A after mixing and low-pass filtering can be expressed as follows:
(3)$$x\left(t\right)=s_{b}(t)+{\tilde{s}}_{\mathrm{int}}(t)+n(t)0\leq t<T_{f},$$
where
(4)$$s_{b}\left(t\right)=\sum_{p=1}^{P}\alpha_{p}e^{j2\pi \left(S\tau_{p}t+f_{0}\tau_{p}\right)}$$
denotes the echo signal of *P* multiple targets, and
(5)$${\tilde{s}}_{\mathrm{int}}(t)=\left[s^{*}\left(t\right)\sum_{q=1}^{Q}s_{\mathrm{int}}\left(t,q\right)\right]*h_{lp}(t)$$
denotes the residual interference signal after mixing and low-pass filtering. Here, $s^{*}\left(t\right)$ represents the complex conjugate of $s\left(t\right)$ used for signal mixing. The *q*th interfering signal is defined as $s_{\mathrm{int}}\left(t,q\right)=s_{q}(t-\tau_{d}(t))$, where $\tau_{d}(t)$ denotes a time delay function composed of delay and time drift. Then, $h_{lp}(t)$ denotes a low-pass filter with a threshold of bandwidth (BW). If the mixed frequency is not within the BW of the low-pass filter, the interference signal does not affect radar A. Finally, n(t) indicates the system noise and measurement error.

If the transmitted signal of aggressor radar B is consistent with that of radar A, the output result ${\tilde{s}}_{\mathrm{int}}\left(t\right)$ is similar to the target after mixing and low-pass filtering. As depicted in Figure 3, we cannot effectively distinguish the target and interference from the 2D-FFT results of $x\left(t\right)$.

The IF signals $x\left(t,m\right),m=0,1,\ldots ,M-1$ of all chirps in a frame after mixing processing are sampled at *N* points to obtain the sampling signal of x(n,m)=x(nΔt,m),n=0,1,…,N−1, and the sampling rate is f_s_, the sampled signal is processed by 1D-FFT in the range dimension, and the corresponding 1D-FFT result is as follows:(6)$$X\left(k,m\right)=\sum_{n=0}^{N-1}x\left(n,m\right)e^{-j2\pi \frac{nk}{N}},$$
where k=0,1,2…N−1.

Because of signal sampling and the existence of time delay functions, the frequency difference between the interference and transmitted signals of the victim radar is constantly changing. The time delay function is determined by the delay and time drift, where the delay represents the time difference between the interference signal and the victim radar. Due to the delay, in the 1D-FFT domains, the peaks introduced by the interference radar are absent in the first or last few chirps within one frame. The time drift affects the alignment of the two radar clocks, which changes the frequency difference between the two signals, and η represents the time drift rate. When the frequency difference between two signals exceeds the frequency resolution $\Delta f=f_{s}/N$ within a continuous frame of time, where $f_{s}$ is the sampling rate of the FMCW radar, a change in the target frequency will be detected in the 1D-FFT result. At this time, the interference signal in the 1D-FFT result behaves similarly to a fast-moving object. Additionally, the critical time drift rate $\eta_{0}$ is as follows:(7)$$\eta_{0}=\frac{f_{s}}{NMT_{c}S}$$

Assume that the clocks of the two radars are aligned at a certain moment, as depicted in Figure 4a, and the first frequency difference between the two signals after mixing is 0. Due to the time drift, the frequency difference between the two signals keeps increasing, as depicted in Figure 4b; when the frequency difference exceeds BW (blue dotted line in Figure 4), the interfering signal will not affect radar A because of the low-pass filter. Over time, as shown in Figure 4c, the interfering signal aligns with the later chirp, the frequency difference between the two signals slowly decreases within the range of the low-pass filter, and the interfering signal again affects radar A. As expressed in (2), there is a long idle time for the radar after sending all chirp signals in one frame, as shown in Figure 4d. Over time, if all chirp signals emitted by radar B are aligned with the idle time of radar A, the interfering signal will not affect the victim radar.

In Figure 5, the 1D-FFT and target detection results under two interference scenarios $\eta \leq \eta_{0}$ and $\eta >\eta_{0}$ are shown, respectively. The figure shows stationary targets, moving targets, and two types of interference. The x-axis represents the chirp number, and the y-axis represents the range. Figure 5a,b is a decibel plot, and its specific dB values can be referred to the color bar in Figure 3. Figure 5c,d is a binary plot, where 1 represents the presence of the target and 0 represents the absence of the target. Figure 5a shows the 1D-FFT result of a delayed signal in one frame when $\eta <\eta_{0}$. For $\eta >\eta_{0}$, the frequency difference between the two signals exceeds Δf, as shown in the 1D-FFT result in Figure 5b, and we can observe that the frequency of the interfering signal has changed. Next, we will design the ID scheme according to the property of the interference signal shown in Figure 5.

## 3. Interference Detection and Mitigation Method

The ability to accurately detect interference is a key factor in effectively suppressing it. Based on the analysis of the different distribution characteristics of targets and interferences in the 1D-FFT results, targets appear as peaks at the same frequency point for all chirps in a frame, the delay appears as the absence of target peaks in the first or last few chirps within one frame, and the time drift appears as a deviation in the peak position of the target in different chirps. Therefore, by applying a 1D-CFAR detector to the 1D-FFT results and summing the target results in the range dimension, interference can be detected based on the number of chirps in which the target exists and then is suppressed. The flowchart of the proposed interference detection and mitigation method is shown in Figure 6. The algorithm can be divided into three steps: target detection, interference detection, and interference mitigation.

### 3.1. Target Detection

A cell-averaging CFAR (CA-CFAR) detector [25] is used for target detection, which detects the target peak in the 1D-FFT result and sets a threshold G to limit the target peak. The selection of reference cells, protected cells, the false alarm rate, etc., can be based on a specific scenario. Applying the CA-CFAR detector to each chirp results in a target matrix B(k, m), whose size is the chirp number M multiplied by sampling point number N, as depicted in Figure 5c,d. In each chip, the targets detected by the CFAR are represented by yellow dots. In Figure 5c, there is a target absence for the interference signal, and in Figure 5d, the targets introduced by the interference appear at different frequency points.

Compared with the normal targets in the 1D-FFT results, the interfering targets show different properties because of the time drift between the victim radar and the aggressor radar. The differences between the target and the interference prompt us to propose the interference detection and mitigation method, as shown in Algorithm 1.

**Algorithm 1.** IM Method Based on Time-domain Signal Reconstruction**Data:** complex signal x(n,m) after mixing 
**Results:** complex signal x′(n,m) after IM Begin  Xk,m=FFT(x(n,m));[N,M]=size(X(k,m)); B=CA−CFARDetector(X);For k0 = 1 to N  If ∑m=0M−1Bk0,m+max∑m=0M−1Bk0+1,m,∑m=0M−1Bk0−1,m<KFor m0 = 1 to M If Bk0,m0==1Ik0,m0=1;  xi(n,m)=TimeReconstruction(X,x);i=i+1; End  End End  xIF(n,m)=∑i=1Cxi(n,m); x′(n,m)=x(n,m)−xIF(n,m); End End

### 3.2. Interference Detection

After the above processing, the target result *B*(*k*, *m*) is detected in the range dimension. For the current frequency point $k_{0}$, the ID formula is as follows:(8)$$\sum_{m=0}^{M-1}B\left(k_{0},m\right)+\max\left(\sum_{m=0}^{M-1}B\left(k_{0}+1,m\right),\sum_{m=0}^{M-1}B\left(k_{0}-1,m\right)\right)<K$$
where K is an empirical parameter with values ranging from 0 to *M*, and max() represents the maximum value of the content in parentheses. If the number of targets at the frequency point satisfies (8), it is judged that the frequency point is an interference frequency point; otherwise, it is a target. Considering that the target may jitter in the range dimension, we consider the number of targets in the adjacent range dimension when calculating the number of targets.

After detection in the Doppler dimension, an ID result *I(k, m)* with the same size as the target detection result is obtained.

### 3.3. Interference Mitigation

After obtaining the interference result *I*(*k*, *m*) through the above steps, we suppress the interference. Specifically, assuming that the total number of interference points is C, i=0,1…C, we traverse all interference points $\left(k_{i},m_{i}\right)$ in turn to reconstruct the interference signal. According to the 1D-FFT results of the chirp where the interference point is located, the interference signal $x_{i}(n,m)$ in the time domain is reconstructed. Now, the time-domain interference reconstruction (TIR) scheme is detailed below.

The interference frequency is calculated by
(9)$$f_{i}=(k_{i}+\Delta k_{i})\frac{f_{s}}{N},$$
where $\Delta k_{i}$ denotes the index deviation as follows:(10)$$\Delta k_{i}=\left\{\begin{array}{cc} \frac{\left|X\left(k_{i}\;+\;1,m_{i}\right)\right|}{\left|X\left(k_{i}\;+\;1,m_{i}\right)\right|\;+\;\left|X\left(k_{i},m_{i}\right)\right|} & \left|X\left(k_{i}+1,m_{i}\right)\right|>\left|X\left(k_{i}-1,m_{i}\right)\right|\\ -\frac{\left|X\left(k_{i}\;-\;1,m_{i}\right)\right|}{\left|X\left(k_{i}\;-\;1,m_{i}\right)\right|\;+\;\left|X\left(k_{i},m_{i}\right)\right|} & \left|X\left(k_{i}\;+\;1,m_{i}\right)\right|<\left|X\left(k_{i}\;-\;1,m_{i}\right)\right|\\ 0 & \left|X\left(k_{i}\;+\;1,m_{i}\right)\right|=\left|X\left(k_{i}\;-\;1,m_{i}\right)\right| \end{array}\right.$$
which is calculated by the 1D-FFT results at the interference point and the two adjacent points.

The amplitude $A_{i}$ and phase $\varphi_{i}$ of the interference signal can be obtained using
(11)$$\begin{array}{l} A_{i}=\frac{1}{N}\left|\sum_{n=0}^{N-1}x(n,m_{i})e^{-j2\pi f_{i}n}\right|\\ \varphi_{i}=Arg\left(\sum_{n=0}^{N-1}x(n,m_{i})e^{-j2\pi f_{i}n}\right), \end{array}$$
where | | represents the amplitude of the complex number, and *Arg*() represents the phase of the complex number.

According to $f_{i}$, $A_{i}$, and $\varphi_{i}$, the interference signal $x_{i}(n,m)$ corresponding to the frequency point in its chirp is reconstructed.
(12)$$x_{i}(n,m)=A_{i}e^{j(2\pi f_{i}n+\varphi_{i})}n=0,1\ldots N-1$$

After reconstructing all interference signals, obtain the sum of all interference signals $x_{IF}(n,m)$. The time-domain signals $x^{\prime }(m,n)$ after IM are, respectively,
(13)$$x_{IF}(n,m)=\sum_{i=1}^{C}x_{i}(n,m),$$
(14)$$x^{\prime }(n,m)=x(n,m)-x_{IF}(n,m),$$
where m=0,1…M−1,n=0,1…N−1.

## 4. Numerical Simulations

The numerical simulation results validate the IM performance of the proposed method. Meanwhile, the results are compared with the two commonly used IM methods in CFAR: the zeroing method (ZM) and the amplitude correction method (AC) [16].

### 4.1. Performance Metrics

To facilitate the comparison of different IM methods and quantitatively evaluate the accuracy of each method in recovering the target signal, the signal-to-interference-plus-noise ratio (SINR) and correlation coefficient ρ of the signal relative to the clean reference signal are used as performance metrics. The SINR and ρ are defined as follows [26]:(15)$$\mathrm{SINR}(s_{b},\hat{s})=20\log_{10}\frac{{\left\|s_{b}\right\|}_{2}}{{\left\|\hat{s}-s_{b}\right\|}_{2}},$$
(16)$$\rho (s_{b},\hat{s})=\frac{{\hat{s}}^{H}s_{b}}{{\left\|s_{b}\right\|}_{2}{\left\|\hat{s}\right\|}_{2}},$$
where $s_{b}$ and $\hat{s}$ denote the vectors of the reference signal and the reconstructed signal after IM, respectively, ${\hat{s}}^{H}$ represents the conjugate transpose of $\hat{s}$, and ${\left\|\;\right\|}_{2}$ is the 2-norm of a vector.

### 4.2. Simulation Experiment

To better describe the interference situation in question, we built a simulation scenario in the Windows 11 system using MATLAB R2018b software. In this simulation scenario, two targets, T1 and T2, two invasion radars, B1 and B2, and one victim radar A are set. The simulation radar parameters are listed in Table 1, and the simulation target and radar position parameters are listed in Table 2. According to the parameters in Table 1, we can obtain $\eta_{0}=2.91\times 10^{-7}$ according to (7) and the maximum applicable speed $v_{\max}=c/2BMT_{c}=43.62\;m/s$ for this method, where $B=SN/f_{s}$ and c is the speed of light. When the target’s speed is less than $v_{\max}$, there will be no change in target frequency in the 1DFFT result. In this simulation scenario, a normal driving vehicle will not cause a change in the target frequency. All parameters in Table 2 are relative to radar A, and the velocity is selected as the positive direction in the direction close to radar A. To better demonstrate the IM ability of the proposed method, the delay and time drift are simulated in one scenario. Radar B1 only has a delay $C_{o}=30T_{c}=1.05\;\mathrm{ms}$ and no time drift; radar B2 only has a time drift $\eta =2\times 10^{-6}$ and no delay. Considering the thermal noise and measurement errors of the radar system, complex white Gaussian noise with an SNR of 10 dB is added to the simulation scenario.

In the simulation scenario, Figure 7a shows the time-domain signal at the 40th chirp. According to the proposed method, first, 256-point 1D-FFT is performed on the signal in each chirp, that is, the range dimension to obtain 1D-FFT results, as depicted in Figure 7b. From the figure, there is a horizontal straight line at 40.26 m and 22.28 m, which are the stationary and moving targets, respectively; there is a missing straight line at 12.12 m; and there is a straight line with five times of peak position offset near 6.45 m, which are two interference signals. Then, 128-point FFT is performed on the Doppler dimension for 1D-FFT results to obtain 2D-FFT results, as depicted in Figure 7c. From the figure, there are four peaks corresponding to four target signals. From the 2D-FFT result graph, if the distance and velocity parameters of the target are unknown in advance, the target and interference cannot be effectively distinguished. After the 1D-FFT spectrum is detected by CA-CFAR, the target results required for interference detection can be obtained, as depicted in Figure 7d. The number of reference cells in the CA-CFAR detector is nine; the number of protective cells is five; the false alarm rate is $10^{-6}$; and the threshold is 40 dB. Then, the target results $B\left(m,n^{\prime }\right)$ are summed in the range dimension to obtain the number of targets at each frequency point. According to the threshold K=124, the corresponding frequency peaks of all interference signals are obtained. After obtaining the frequency peaks of the interference signal, the reconstruction of the interference signals is completed according to (9)–(12). Then, according to the chirp numbers of the interference signals, the corresponding interference signals are eliminated.

Figure 7e shows the time-domain signal at the 40th chirp after IM. The clear signal can be transformed using 1D-FFT, CA-CFAR detector, and 2D-FFT to observe the effect of IM. The 1D-FFT result without interference is shown in Figure 7f; the target result is shown in Figure 7g; and the 2D-FFT result is shown in Figure 7h. In Figure 7f–h, the interferences at 12.12 m and 6.45 m have been suppressed, indicating that the IM effect has been achieved. In Figure 7c,h, it can be seen that the amplitude of the interference signal is reduced by approximately 25 dB.

### 4.3. Effect of SNR on Interference Mitigation

Due to the noise in the collected signal, the target detection effect is affected, which in turn affects the IM effect of the proposed method. In this section, we use the same targets and interferences as in Section 4.2 and change the noise level to investigate the changes in the IM ability of the three methods at different SNRs.

We consider noise levels ranging from −20 to 20 dB and perform 1000 Monte Carlo runs at each noise level. The performance indicators of the three IM methods are depicted in Figure 8. The bottom and top of each rectangular box in the figure represent the 25th and 75th percentiles of the samples, respectively, and the lines extending above and below the rectangular box represent the range between the maximum and minimum values of the samples. As depicted in Figure 8a, the profile of the TIR method shows an approximately straight line, with the SINR continuously increasing as the SNR increases. However, when SNR ≤ 0 dB, the ZM and AC methods can maintain an increase in SINR. When SNR > 0 dB, the increase in SINR becomes slower and slower as SNR increases. When SNR = 15 dB, the SINR obtained by the TIR method is 10 dB higher than that obtained by the ZM and AC methods. In the current simulation environment, the TIR method obtains a better SINR than the ZM and AC methods when SNR > 0 dB. When SNR ≤ 0 dB, the TIR method also obtains an SINR similar to the two comparison methods. The target detection effect is better than those of the ZM and AC methods at high SNRs due to the low noise intensity. The ZM method is the simplest method to suppress interference; however, although it can suppress interference, it also removes useful signals, resulting in the risk of missed alarms. With regard to the AC method, although the adjacent FFT results are used for spectrum recovery, its effect is slightly better than that of the ZM method under high SNR conditions. Finally, using the TIR method, under high SNR conditions, interference can be accurately detected, and the frequency, amplitude, and phase of the interference target can be accurately estimated so that the interference can be accurately removed in the time domain, and useful signals can be maximally preserved.

Figure 8b shows that when the SNR changes from −20 t to 20 dB, the magnitude of correlation coefficients of the TIR method increases from 0.1 to 1, while with the ZM and AC methods it can only increase from 0.1 to 0.7 and 0.8, respectively, and is almost stable under high SNR conditions. We can conclude that at high SNRs, the amplitude of the signal correlation coefficient obtained by the TIR method is higher than those of the other two methods, and is almost the same as the other two methods at low SNRs. Figure 8c shows that as the input SNR increases, the phase of the correlation coefficient of the three methods gradually tends to zero. Therefore, in terms of the evaluation indicators, the proposed method is superior to the other two methods.

## 5. Experimental Results

This section presents experimental results to verify the performance of the proposed method. Two experimental scenarios were set up: one was a direct interference experiment to verify the correctness of the simulation results, and the other was a reflection interference experiment to verify the authenticity of the proposed research background. In the experimental part, two Texas Instruments IWR1843 radar boards were used as the victim radar A and attacker radar B. Radar data were collected using the DCA1000EVM data collection board. The radar board burning software was UniFlash6.1.0, and the radar board configuration software was mmWave Studio 02.01.00.00.

### 5.1. Direct Interference Experiment

The experimental scenario is depicted in Figure 9. The parameter settings of the two radar boards are the same, as listed in Table 3. The position parameters between the two radar boards and the corner reflector are listed in Table 4. The horizontal angle in Table 4 is measured clockwise based on the connecting line between the victim radar and the stationary target, while the elevation angle is measured clockwise based on the horizontal direction of the victim radar, and downward is positive. For the parameter settings of radar A, we can obtain $\eta_{0}=2.91\times 10^{-7}$ according to (7).

According to the proposed method, the first step is to extract the signal collected by radar A. The signal collected by radar A is transformed from the time domain to the frequency domain through 1D-FFT, and the frequency spectrum is shown in Figure 10a. The figure shows interference as a diagonal line near 9.77 m and a stationary target as a straight line near 3.32 m. After CFAR detection and threshold selection (threshold G=95 dB), the target detection result is illustrated in Figure 10b. The 2D-FFT result of the signal is shown in Figure 10c, showing a large amplitude environment noise floor elevation next to the stationary target. The peak value at the range of 0 m and speed of 0 m/s in the figure is the interference of the radar board itself, which can be ignored in the set scenario. In Figure 10b, it can be seen that the interference lasts from the 1st chirp to the 55th chirp, and the frequency point changes from the 44th to the 56th. The time drift $\eta_{1}=f_{s}/m_{1}NT_{c}S=7.11\times 10^{-6}$ can be obtained in this scenario, where $m_{0}=(55-1+1)/$(56−44)=4.58 represents the number of chirps required for the target frequency to appear offset. Then, the target result is summed up in the range dimension to obtain the number of targets at each frequency point. The frequency points corresponding to all interference signals are found based on the threshold K=124. After obtaining the frequency points of the interference signal, the reconstruction of the interference signal is completed according to (9)–(12). Then, the interference is removed separately according to the chirps where it is located.

At this time, the 1D-FFT and target results are re-obtained, as depicted in Figure 10d,e. In the figures, the interference signal at 9.77 m is suppressed, and the stationary target at 3.32 m is preserved. The 2D-FFT results are shown in Figure 10f, where there is no significant change in the peak value of the stationary target, and the interference only leaves a portion of the signal that can be considered noise.

The direct interference experiment verifies the rationality of the simulation experiment and also proves the effectiveness of the proposed algorithm in detecting and suppressing interference. From Figure 9c,f it can be seen that the amplitude of the interference signal is reduced by approximately 20 dB, which provides a good foundation for subsequent data processing.

### 5.2. Reflection Interference Experiment

The experimental results presented in this section verify the performance of the proposed method in a real environment. The experimental scenario is shown in Figure 11. A stationary car is parked in front of the radar boards. This creates a simple road experiment scenario. The parameter settings of the two radar boards are the same as listed in Table 5. The position parameters between the two radar boards and the vehicle are listed in Table 6. The horizontal angle in Table 6 is measured clockwise based on the connecting line between the victim radar and the stationary target, while the elevation angle is measured clockwise based on the horizontal direction of the victim radar, and downward is positive. For the parameter settings of radar A, $\eta_{0}=2.91\times 10^{-7}$ according to (7).

According to the proposed method, the first step is to extract the signal collected by radar A. The signal collected by radar A is transformed from the time domain to the frequency domain through 1D-FFT, and the frequency spectrum is shown in Figure 12a. The figure shows interference as a diagonal line near 5.08 m and a stationary target as a straight line at 1.95 m. After CFAR detection and threshold selection (threshold G=50 dB), the target detection result is illustrated in Figure 12b. The 2D-FFT result of the signal is shown in Figure 12c, showing a large amplitude environment noise floor elevation next to the stationary target. In Figure 12b, it can be seen that the interference lasts from the 1st chirp to the 74th chirp and the frequency point changes from the 23rd to the 35th. The time drift $\eta_{2}=f_{s}/m_{2}NT_{c}S=6.04\times 10^{-6}$ can be obtained in this scenario, where $m_{2}=(74-1+1)/(35-23)=6.17$. Then, the target result is summed up in the range dimension to obtain the number of targets at each frequency point. The frequency points corresponding to all interference signals are found based on the threshold K=124. After obtaining the frequency points of the interference signal, the reconstruction of the interference signal is completed according to (9)–(12). Then, the interference is mitigated according to the chirps where it is located.

At this time, the 1D-FFT and target results are re-obtained, as depicted in Figure 12d,e. In the figures, the interference signal at 5.08 m is suppressed, and the stationary target at 1.95 m is preserved. The 2D-FFT result is shown in Figure 12f, where the peak value of the stationary target does not change considerably. Although a portion of the signal remains after suppressing interference, it is sufficient to be considered noise and will not affect the target signal.

The reflection interference experiment proves the existence of the proposed interference scenario in real life. From Figure 12c,f, it can be seen that the proposed method can reduce the amplitude of the interference signal by about 25 dB in this scenario.

## 6. Conclusions and Outlook

### 6.1. Conclusions

This study proposes a method for interference detection and mitigation based on 1D-FFT results for FMCW radar systems. The proposed method uses the difference in 1D-FFT results between targets and interferences. Then, by reconstructing the interference signal in the time domain using the information about the interference from the 1D-FFT results, the interference signal can be removed from the time domain without affecting the target signal. In the simulation and experimental results, the detection and mitigation of interference targets showed good results. No interferences were observed in the 2D-FFT results of both after IM. Subsequently, this method was compared with the ZM and AC methods. Under high SNR conditions, the SINR of the TIR method increases linearly with the increase in the SNR, while the increased speed of the ZM and AC methods is significantly slower than that of the TIR method. When SNR = 15 dB, the SINR obtained by the TIR method is 10 dB higher than that obtained by the ZM and AC methods The magnitude of the correlation coefficient of the TIR method can approach 1, while with the ZM and AC methods it can only stabilize at 0.7 and 0.8, respectively. In simulation scenarios, this method can achieve an interference suppression effect of 25 dB. In direct interference experiments, this method can achieve a 20 dB interference suppression effect, and in reflection interference experiments, this method can achieve a 25 dB interference suppression effect.

### 6.2. Outlook

The methods outlined in this article still have some problems that can be further studied. In the simulation and experiments presented in this article, a maximum of two targets and two interferences are considered. In further work, we will add more interferences and different types of targets in the simulation, and add more experimental scenarios in the experiment, constantly approaching the actual road situation. We only conducted numerical analysis on the performance indicators SINR and ρ mentioned in the paper, and we will investigate the performance indicators theoretically in the future. Moreover, we will discuss more performance indicators, such as the gain in saving computational resources, implementation costs, processing delay, computational complexity, and detection accuracy.

## Figures and Tables

**Figure 1 sensors-23-07113-f001:**
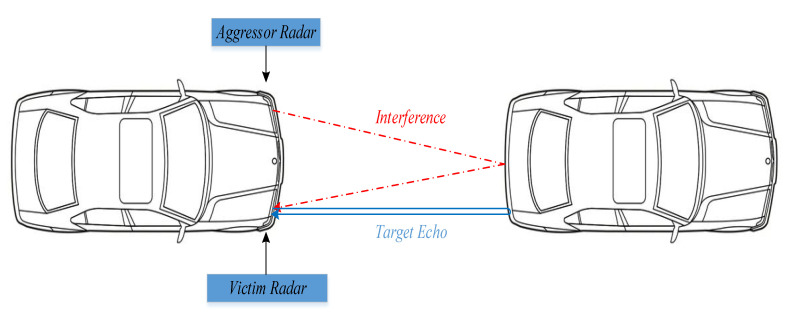
False target interference scenario.

**Figure 2 sensors-23-07113-f002:**
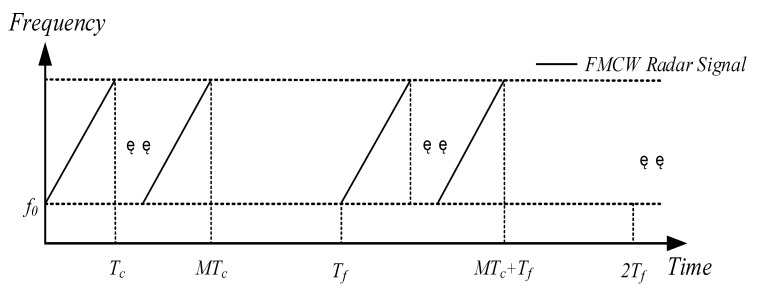
Schematic of an FMCW radar signal.

**Figure 3 sensors-23-07113-f003:**
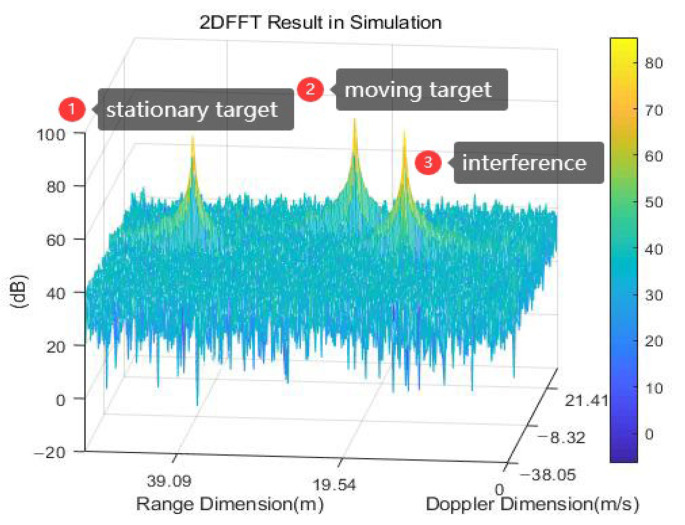
The 2D-FFT result diagram of $x\left(t\right)$ (including moving target, stationary target, and interfering signal).

**Figure 4 sensors-23-07113-f004:**
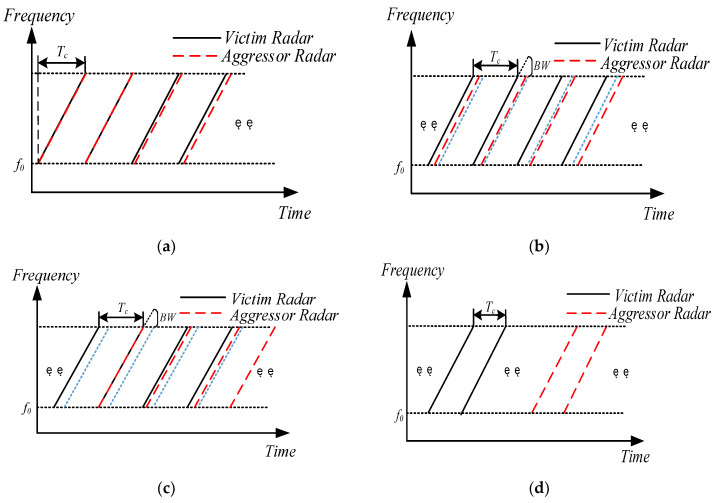
Schematic of aggressor and victim radar signals when $\eta \leq \eta_{0}$: (**a**) shows that the clocks of the aggressor and victim radars are aligned at the beginning; (**b**) shows that the frequency difference between the aggressor and victim radar signals exceeds BW; (**c**) shows that the chirp of the aggressor radar signal is aligned with the idle time of the victim radar; (**d**) shows that all chirps of the aggressor radar signal are aligned with the idle time of the victim radar.

**Figure 5 sensors-23-07113-f005:**
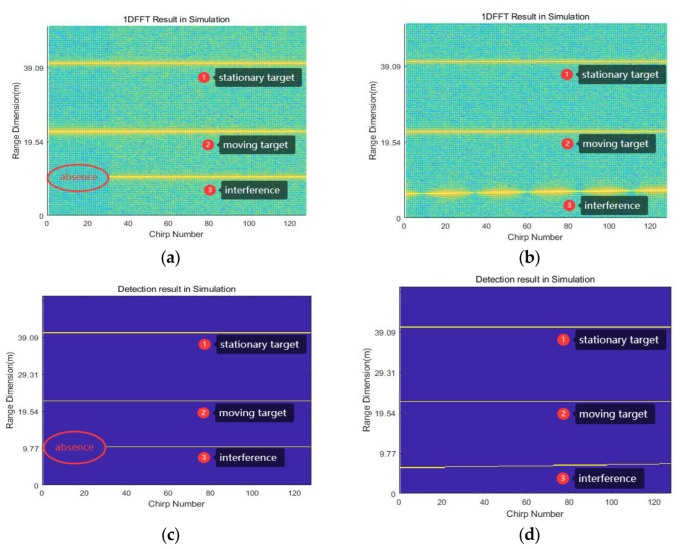
The 1D-FFT result and the target detection result of the signal $x\left(t\right)$: (**a**,**c**) represent the result when $\eta \leq \eta_{0}$; (**b**,**d**) represent the result when $\eta >\eta_{0}$.

**Figure 6 sensors-23-07113-f006:**
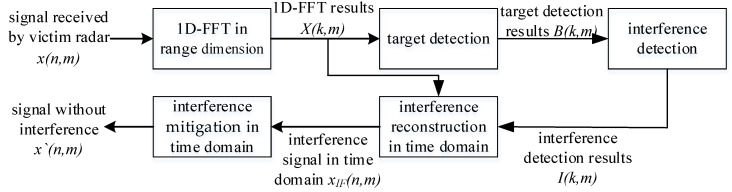
Flowchart of the proposed IM method.

**Figure 7 sensors-23-07113-f007:**
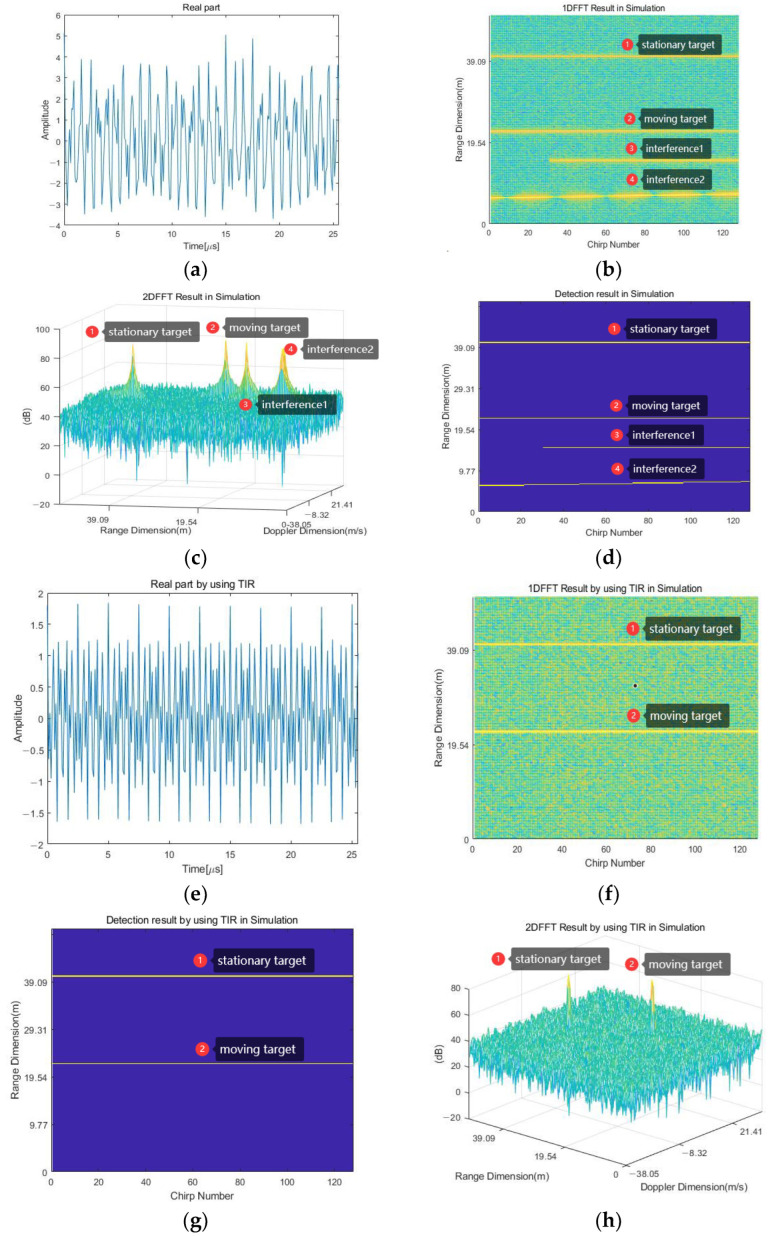
Simulation results. (**a**) Time-domain diagram at the 40th chirp before IM. (**b**) The 1D-FFT result before IM. (**c**) Target result before IM. (**d**) The 2D-FFT result before IM. (**e**) Time-domain diagram at the 40th chirp after IM. (**f**) The 1D-FFT result after IM. (**g**) Target result after IM. (**h**) The 2D-FFT result after IM.

**Figure 8 sensors-23-07113-f008:**
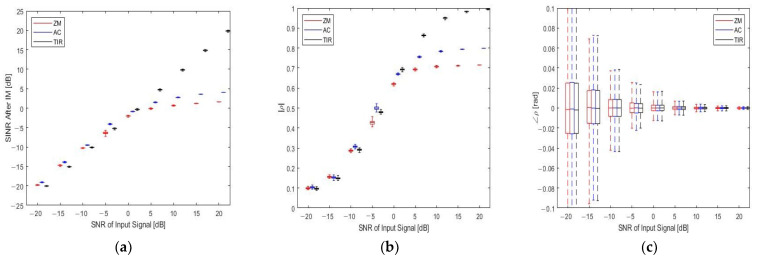
Quantitative comparison of the interference mitigation performance of the TIR, ZM, and AC methods under different input signal SNR conditions. (**a**–**c**) show the variations in SINR, the magnitudes, and phase angles of correlation coefficients of the recovered beat signals after IM, respectively.

**Figure 9 sensors-23-07113-f009:**
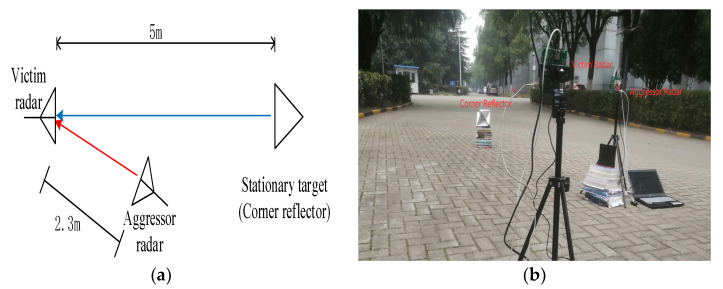
Direct interference experimental scenario: (**a**) schematic and (**b**) physical diagram.

**Figure 10 sensors-23-07113-f010:**
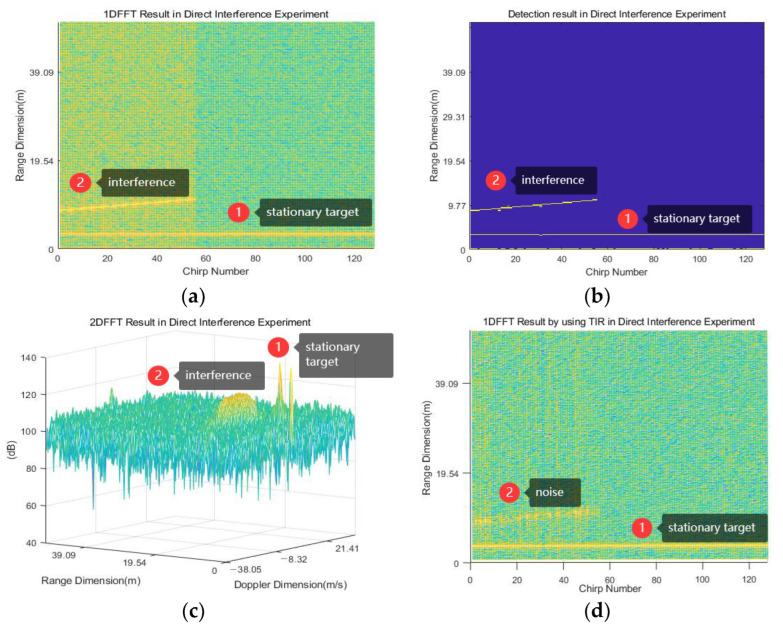
Direct interference experimental results: (**a**–**c**) represent the 1D-FFT, 2D-FFT, and target results before IM, respectively; (**d**–**f**) represent the 1D-FFT, 2D-FFT, and target results after IM, respectively.

**Figure 11 sensors-23-07113-f011:**
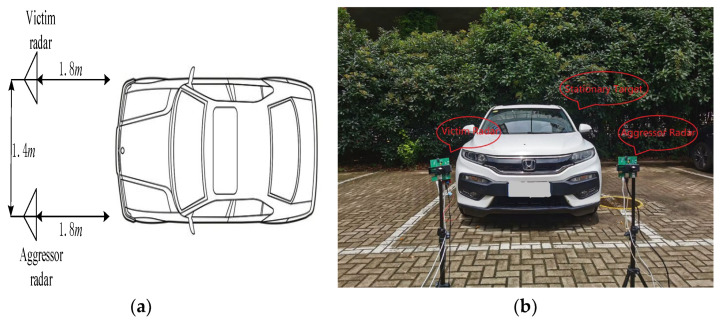
Reflection interference experimental scenario: (**a**) schematic and (**b**) physical diagram.

**Figure 12 sensors-23-07113-f012:**
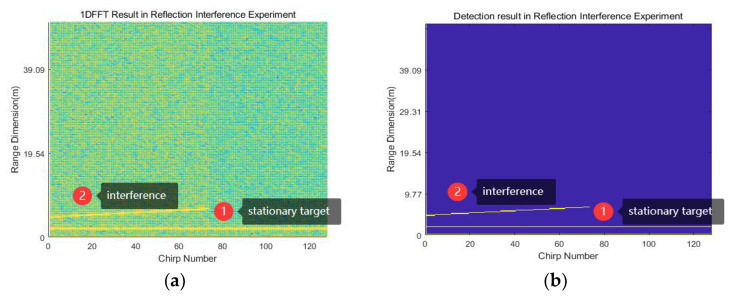
Reflection interference experimental results: (**a**–**c**) represent the 1D-FFT, 2D-FFT, and target results before IM, respectively; (**d**–**f**) represent the 1D-FFT, 2D-FFT, and target results after IM, respectively.

**Table 1 sensors-23-07113-t001:** Simulated radar parameters.

Parameter	A	B1	B2
Starting frequency (f_0_) (GHz)	77	77	77
Modulation slope (*S*) (MHz/μs)	29.982	29.982	29.982
Duration of one chirp (*T_c_*) (μs)	35	35	35
Duration of one frame (*T_f_*) (ms)	50	50	50
Sampling rate (*f_s_*) (MHz)	12.5	10	10
Sampling points (*N*)	256	256	256
Number of chirps in one frame (*M*)	128	128	128
Number of simulation frames	1000	1000	1000

**Table 2 sensors-23-07113-t002:** Target and radar position parameters in the simulation.

Parameter	T1	T2	B1	B2
Distance (m)	40	22	6	15
Speed (m/s)	0	15	0	0
Echo amplitude	0.8	1	2.1	1.3

**Table 3 sensors-23-07113-t003:** IWR1843 Radar board setting parameters in the direct interference experiment.

Parameter	A	B
Starting frequency (*f*_0_) (GHz)	77	77
Modulation slope (*S*) (MHz/μs)	29.982	29.982
Duration of one chirp (*T_c_*) (μs)	35	35
Duration of one frame (*T_f_*) (ms)	50	50
Sampling rate (*f_s_*) (MHz)	10	10
Sampling points (*N*)	256	256
Number of chirps in one frame (*M*)	128	128
Number of frames	1000	1000

**Table 4 sensors-23-07113-t004:** IWR1843 Radar board positional parameters in the direct interference experiment.

Parameter	Value
Distance between two radars (m)	2.3
Linear distance from radars to target (m)	5.0
Distance from radars to ground (m)	1.7
Azimuth between the victim radar and stationary target (°)	0
Azimuth between the victim radar and aggressor radar (°)	20
Elevation angle between the victim radar and stationary target (°)	13.5
Elevation angle between the victim radar and aggressor radar (°)	0

**Table 5 sensors-23-07113-t005:** IWR1843 Radar board setting parameters in the reflection interference experiment.

Parameter	A	B
Starting frequency (*f*_0_) (GHz)	77	77
Modulation slope (*S*) (MHz/μs)	29.982	29.982
Duration of one chirp (*T_c_*) (μs)	35	35
Duration of one frame (*T_f_*) (ms)	50	50
Sampling rate (*f_s_*) (MHz)	10	10
Sampling points (*N*)	256	256
Number of chirps in one frame (*M*)	128	128
Number of frames	1000	1000

**Table 6 sensors-23-07113-t006:** IWR1843 Radar board positional parameters in the reflection interference experiment.

Parameter	Value
Distance between two radars (m)	1.4
Linear distance from radars to vehicle (m)	1.8
Distance from radar to ground (m)	1.0
Azimuth between the victim radar and stationary target (°)	0
Azimuth between the victim radar and aggressor radar (°)	90
Elevation angle between the victim radar and stationary target (°)	0
Elevation angle between the victim radar and aggressor radar (°)	0

## Data Availability

Data are available upon request.

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
