# Peer review of "FMCW Radar System Interference Mitigation Based on Time-Domain Signal Reconstruction"

_sensors, 2023, doi:10.3390/s23167113_

Round 1
Reviewer 1 Report
The article titled, “FMCW Radar System Interference Mitigation Based on Time Domain Signal Reconstruction” was submitted by Xu et al in the Sensors Journal. This study proposes an interference mitigation method for continuous wave radar systems using time-domain signal reconstruction. It identifies interferences through differences in 1D-FFT results compared to target signals and removes them without affecting the target signal. The method is shown to be effective in both simulations and experiments.
The paper is interesting and contains good results. However, I have the following comments and questions which need to be answered before the final decision.
· What is the basis of the proposed interference mitigation method for frequency-modulated continuous wave radar systems?
· How does the difference in one-dimensional fast Fourier transform (1D-FFT) result between targets and interferences enable interference removal in the proposed method?
· What are the different distribution characteristics of targets and interferences in the 1D-FFT results, and how do they enable interference detection in the proposed method?
· Could you explain the three steps of the proposed interference detection and mitigation algorithm (Algorithm 1)? How does it effectively suppress interference based on the analysis of 1D-FFT results?
· How do the three-interference mitigation (IM) methods perform in terms of Signal-to-Interference-plus-Noise Ratio (SINR) with respect to varying input Signal-to-Noise Ratio (SNR)? Which method shows better SINR results under different SNR conditions?
· What advantages does the TIR (Time-domain Interference Reconstruction) method offer over the ZM (Zero-padding Method) and AC (Adjacent Chirp) methods in interference suppression under high SNR conditions? How does the TIR method accurately detect and remove interference in the time domain, preserving useful signals maximally?
Reviewer 2 Report
Contributions must be clarified and expanded in the introduction section. Please, enumerate these.
In Figs. add scales for each axis. For instance, range dimension is in ... meters. In figure 3, the z scale means... Please, review all figures.
The parameters of tables I y II must be justified.
A comparison among simulated and experimental results must be added. At the same time, a comparison with other methods is not presented, please insert it.
Future works should be added with details in the conclusion section.
Abbreviations at the end of the paper must be inserted, please see MDPI template.
Minor editing of English language required
Reviewer 3 Report
In this manuscript, the so called FMCW Radar System Interference Mitigation Based on Time- Domain Signal Reconstruction is analyzed and discussed.
Nevertheless, the commented key issues must be re emphasized to the authors, due to the low content quality of this eidtion.
(1) Although 21 references are included in this work, but more than11 references are published more than 4 years, even 5 years, which definitely
could not refect the edge cutting develoment of this reserach direction of FMCW Radar System Interference Mitigation.
(2) In the abstract, the gain in saving computation resources, implementation cost, processing delay, computaion complexity and impoving the detection&ranging accuracy must be explicitly provided. The numerical
improvement in FMCW Radar System Interference Mitigation must be included as well.
(3) The too apparent similarity is derived between the abstract and the conclusion. Frankly, the conclusion is just the copy of the abstract. The several distinct numerical results and performance metrics must be presented
in the conclusion, compared to the abstract.
(4) The flow chart of the proposed FMCW Radar System Interference Mitigation sheme must be included in the work, and the input and output signal of each module must be labeled. And the expression of these input and ouput signals
must be presented.
(5) Many parameters are concerned in this work, but only several parameters are included in Table 1-3. The table of all modules, components and parameters must be included in the modification of this work and the reference and
setting support of all parameters and the datasheets must be labelled to prove the physical implementability and repeatability.
(6) The experimental compoments, platform and operating procedure must be added to this manuscript via several separate figures and statement, for the readers to vertify the repeatability of these so called key numerical experimental results.
Moreover, the parameters of all concernedexperimental compoments, platform must be included in one table for the convenience of the readers to regenerate all these plots results.
(7) The advancement of the proposed scheme is quite doubtful. There is no comparative numerically comparative estimation of the proposed scheme with the edge cutting schemes reported within 3 three years.
The so called comparison with the out of date scheme in this manuscript could not sufficiently vertify the advancement and superiority of the proposed method.
(8) The so called experimental scenario in Fig 8 is far away from the envisioned application scenario in Fig , 1 , especially the spatial positon, azimuth and elevation relationship.
The experiment configuration must be modified to reflect the realistic automotive radar situation as shown in Fig 1.
(9) Moreover, the match extent between the simulation result and the experimental results must be systematic analyzed and numerically discussed to identify the reliability of the simulation works.
Moreover, the simulation paltform and software environment must be clearly illustrated for prove the repeatability.
(10) How many interference how intense interference could be identified and mitigated? The performance limit of the proposed method should be theotrical and numerically investigated.
Before all above modifications are made, I cannot recommend this manuscript to publish.
Careful english checking is still essential for this edition.
Reviewer 4 Report
1. The abstract should be concise about the work presented in the paper.
2. The literature survey is weak and needs to improve by considering new and relevant references.
3. The paper's main contributions should be highlighted more than the existing studies.
4. The readability of the paper should be improved.
5. Equation parameters need to be described properly.
6. The proposed frequency-modulated continuous wave radar systems based on time-domain signal reconstruction design must be described in greater detail in order to be able to validate its performance in the implementation of this proposed system.
7. In general, the work lacks formality in its development, particularly in the presentation of interference detection and mitigation method based on time-domain signal reconstruction system models and algorithms.
8. Make a precise description of the results presented by means of graphs in Figures: 5, 6, 7, and 9. That is, make a description of the results represented for each of the graphs of the previously mentioned figures.
9. The results section should be explained comprehensively with positive points of the proposed work.
10. Report the conclusions taking into account the results data obtained through simulations and experimental tests.
Moderate editing of the English language required
Round 2
Reviewer 1 Report
I am satisfied with the revision.
Reviewer 2 Report
All observations were done, congrats.
Reviewer 4 Report
The authors cleared all my questions and accepted the paper in its present form.
Minor editing of English language required